# Restricting exchangeable nonparametric distributions

**Sinead A. Williamson**
University of Texas at Austin

**Steven N. MacEachern**
The Ohio State University

**Eric P. Xing**
Carnegie Mellon University

## Abstract

Distributions over matrices with exchangeable rows and infinitely many columns are useful in constructing nonparametric latent variable models. However, the distribution implied by such models over the number of features exhibited by each data point may be poorly-suited for many modeling tasks. In this paper, we propose a class of exchangeable nonparametric priors obtained by restricting the domain of existing models. Such models allow us to specify the distribution over the number of features per data point, and can achieve better performance on data sets where the number of features is not well-modeled by the original distribution.

## 1 Introduction

The Indian buffet process [IBP, 11] is one of several distributions over matrices with exchangeable rows and infinitely many columns, only a finite (but random) number of which contain any non-zero entries. Such distributions have proved useful for constructing flexible latent feature models that do not require us to specify the number of latent features *a priori*. In such models, each column of the random matrix corresponds to a latent feature, and each row to a data point. The non-zero elements of a row select the subset of features that contribute to the corresponding data point.

However, distributions such as the IBP make certain assumptions about the structure of the data that may be inappropriate. Specifically, such priors impose distributions on the number of data points that exhibit a given feature, and on the number of features exhibited by a given data point. For example, in the IBP, the number of features exhibited by a data point is marginally Poisson-distributed, and the probability of a data point exhibiting a previously-observed feature is proportional to the number of times that feature has been seen so far.

These distributional assumptions may not be appropriate for many modeling tasks. For example, the IBP has been used to model both text [17] and network [13] data. It is well known that word frequencies in text corpora and degree distributions of networks often exhibit power-law behavior; it seems reasonable to suppose that this behavior would be better captured by models that assume a heavy-tailed distribution over the number of latent features, rather than the Poisson distribution assumed by the IBP and related random matrices.

In certain cases we may instead wish to add constraints on the number of latent features exhibited per data point, particularly in cases where we expect, or desire, the latent features to correspond to interpretable features, or causes, of the data [20]. For example, we might believe that each data point exhibits exactly $S$ features – corresponding perhaps to speakers in a dialog, members of a team, or alleles in a genotype – but be agnostic about the total number of features in our data set. A model that explicitly encodes this prior expectation about the number of features per data point will tend to lead to more interpretable and parsimonious results. Alternatively, we may wish to specify a minimum number of latent features. For example, the IBP has been used to select possible next states in a hidden Markov model [10]. In such a model, we do not expect to see a state that allows *no* transitions (including self-transitions). Nonetheless, because a data point in the IBP can have zero features with non-zero probability, this situation can occur, resulting in an invalid transition distribution.

In this paper, we propose a method for modifying the distribution over the number of non-zero elements per row in arbitrary exchangeable matrices, allowing us to control the number of features per data point in a corresponding latent feature model. We show that our construction yields exchangeable distributions, and present Monte Carlo methods for posterior inference. Our experimental evaluation shows that this approach allows us to incorporate prior beliefs about the number of features per data point into our model, yielding superior modeling performance.

## 2  Exchangeability

We say a finite sequence $(X_1, \ldots, X_N)$ is *exchangeable* [see, for example, 1] if its distribution is unchanged under any permutation $\sigma$ of $\{1, \ldots, N\}$. Further, we say that an infinite sequence $X_1, X_2, \ldots$ is *infinitely exchangeable* if all of its finite subsequences are exchangeable. Such distributions are appropriate when we do not believe the order in which we see our data is important. In such cases, a model whose posterior distribution depends on the order in which we see our data is not justified. In addition, exchangeable models often yield efficient Gibbs samplers.

De Finetti's law tells us that a sequence is exchangeable iff the observations are i.i.d. given some latent distribution. This means that we can write the probability of any exchangeable sequence as

$$P(X_1 = x_1, X_2 = x_2, \ldots) = \int_\Theta \prod_i \mu_\theta(X_i = x_i)\nu(d\theta) \qquad (1)$$

for some probability distribution $\nu$ over parameter space $\Theta$, and some parametrized family $\{\mu_\theta\}_{\theta \in \Theta}$ of conditional probability distributions.

Throughout this paper, we will use the notation $p(x_1, x_2, \ldots) = P(X_1 = x_1, X_2 = x_2, \ldots)$ to represent the joint distribution over an exchangeable sequence $x_1, x_2, \ldots$; $p(x_{n+1}|x_1, \ldots, x_n)$ to represent the associated predictive distribution; and $p(x_1, \ldots, x_n, \theta) := \prod_{i=1}^n \mu_\theta(X_i = x_i)\nu(\theta)$ to represent the joint distribution over the observations and the parameter $\theta$.

### 2.1  Distributions over exchangeable matrices

The Indian buffet process [IBP, 11] is a distribution over binary matrices with exchangeable rows and infinitely many columns. In the de Finetti representation, the mixing distribution $\nu$ is a beta process, the parameter $\theta$ is a countably infinite measure with atom sizes $\pi_k \in (0, 1]$, and the conditional distribution $\mu_\theta$ is a Bernoulli process [17]. The beta process and the Bernoulli process are both *completely random measures* [CRM, 12] – distributions over random measures on some space $\Omega$ that assign independent masses to disjoint subsets of $\Omega$, that can be written in the form $\Gamma = \sum_{k=1}^\infty \pi_k \delta_{\phi_k}$. We can think of each atom of $\theta$ as determining the latent probability for a column of a matrix with infinitely many columns, and the Bernoulli process as sampling binary values for the entries of that column of the matrix. The resulting matrix has a finite number of non-zero entries, with the number of non-zero entries in each row distributed as Poisson($\alpha$) and the total number of non-zero columns in $N$ rows distributed as Poisson($\alpha H_N$), where $H_N$ is the $N$th harmonic number. The number of rows with a non-zero entry for a given column exhibits a "rich gets richer" property – a new row has a one in a given column with probability proportional to the number of times a one has appeared in that column in the preceding rows.

Different patterns of behavior can be obtained with different choices of CRM. A three-parameter extension to the IBP [15] replaces the beta process with a completely random measure called the stable-beta process, which includes the beta process as a special case. The resulting random matrix exhibits power law behavior: the total number of features exhibited in a data set of size $N$ grows as $O(N^s)$ for some $s > 0$, and the number of data points exhibiting each feature also follows a power law. The number of features per data point, however, remains Poisson-distributed. The infinite gamma-Poisson process [iGaP, 18] replaces the beta process with a gamma process, and the Bernoulli process with a Poisson process, to give a distribution over non-negative integer-valued matrices with infinitely many columns and exchangeable rows. In this model, the sum of each row is distributed according to a negative binomial distribution, and the number of non-zero entries in each row is Poisson-distributed. The beta-negative binomial process [21] replaces the Bernoulli process with a negative binomial process to get an alternative distribution over non-negative integer-valued matrices.

# 3 Removing the Poisson assumption

While different choices of CRMs in the de Finetti construction can alter the distribution over the number of data points that exhibit a feature and (in the case of non-binary matrices) the row sums, they retain a marginally Poisson distribution over the number of distinct features exhibited by a given data point. The construction of Caron [4] extends the IBP to allow the number of features in each row to follow a *mixture* of Poissons, by assigning data point-specific parameters that have an effect equivalent to a monotonic transformation on the atom sizes in the underlying beta process; however conditioned on these parameters, the sum of each row is still Poisson-distributed.

This repeatedly occurring Poisson distribution is a direct result of the construction of a binary matrix from a combination of CRMs. To elaborate on this, note that, marginally, the distribution over the value of each element $z_{ik}$ of a row $\mathbf{z}_i$ of the IBP (or a three-parameter IBP) is given by a Bernoulli distribution. Therefore, by the law of rare events, the sum $\sum_k z_{ik}$ is distributed according to a Poisson distribution.

A similar argument applies to integer-valued matrices such as the infinite gamma-Poisson process. Marginally, the distribution over whether an element $z_{ik}$ is greater than zero is given by a Bernoulli distribution, hence the number of non-zero elements, $\sum_k z_{ik} \wedge 1$, is Poisson-distributed. The distribution over the row sum, $\sum_k z_{ik}$, will depend on the choice of CRMs.

It follows that, if we wish to circumvent the requirement of a Poisson (or mixture of Poisson) number of features per data point in an IBP-like model, we must remove the completely random assumption on either the de Finetti mixing distribution or the family of conditional distributions. The remainder of this section discusses how we can obtain arbitrary marginal distributions over the number of features per row by using conditional distributions that are not completely random.

## 3.1 Restricting the family of conditional distributions in the de Finetti representation

Recall from Section 2 that any exchangeable sequence can be represented as a mixture over some family of conditional distributions. The support of this family determines the support of the exchangeable sequence. For example, in the IBP the family of conditional distributions is the Bernoulli process, which has support in $\{0, 1\}^\infty$. A sample from the IBP therefore has support in $\{\{0, 1\}^\infty\}^N$.

We are familiar with the idea of restricting the support of a distribution to a measurable subset. For example, a truncated Gaussian is a Gaussian distribution restricted to a contiguous section of the real line. In general, we can restrict an arbitrary probability distribution $\mu$ with support $\Omega$ to a measurable subset $A \subset \Omega$ by defining $\mu^{|A}(\cdot) := \mu(\cdot)\mathbb{I}(\cdot \in A)/\mu(A)$.

**Theorem 1** (Restricted exchangeable distributions)**.** *We can restrict the support of an exchangeable distribution by restricting the family of conditional distributions $\{\mu_\theta\}_{\theta \in \Theta}$ introduced in Equation 1, to obtain an exchangeable distribution on the restricted space.*

*Proof.* Consider an unrestricted exchangeable model with de Finetti representation $p(x_1, \ldots, x_N, \theta) = \prod_{i=1}^N \mu_\theta(X_i = x_i)\nu(\theta)$. Let $p^{|A}$ be the restriction of $p$ such that $X_i \in A, i = 1, 2, \ldots$, obtained by restricting the family of conditional distributions $\{\mu_\theta\}$ to $\{\mu_\theta^{|A}\}$ as described above. Then

$$p^{|A}(x_1, \ldots, x_N, \theta) = \prod_{i=1}^N \mu_\theta^{|A}(X_i = x_i)\nu(\theta) = \prod_{i=1}^N \frac{\mu_\theta(X_i = x_i)\mathbb{I}(x_i \in A)}{\mu_\theta(X_i \in A)}\nu(\theta)\,,$$

and

$$p^{|A}(x_{N+1}|x_1, \ldots, x_N) \propto \mathbb{I}(x_{N+1} \in A) \int_\Theta \frac{\prod_{i=1}^{N+1} \mu_\theta(X_i = x_i)}{\prod_{i=1}^{N+1} \mu_\theta(X_i \in A)}\nu(d\theta) \tag{2}$$

is an exchangeable sequence by construction, according to de Finetti's law. □

We give three examples of exchangeable matrices where the number of non-zero entries per row is restricted to follow a given distribution. While our focus is on exchangeability of the rows, we note that the following distributions (like their unrestricted counterparts) are invariant under reordering of the columns, and that the resulting matrices are separately exchangeable [2].

**Example 1** (Restriction of the IBP to a fixed number of non-zero entries per row)**.** The family of conditional distributions in the IBP is given by the Bernoulli process. We can restrict the support

of the Bernoulli process to an arbitrary measurable subset $A \subset \{0,1\}^\infty$ – for example, the set of all vectors $\mathbf{z} \in \{0,1\}^\infty$ such that $\sum_k z_k = S$ for some integer $S$. The conditional distribution of a matrix $\mathbf{Z} = \{\mathbf{z}_1, \ldots, \mathbf{z}_N\}$ under such a distribution is given by:

$$
\begin{aligned}
\mu_B^{|S}(Z = \mathbf{Z}) &= \frac{\prod_{i=1}^N \mu_B(Z_i = \mathbf{z}_i)\mathbb{I}(\sum_k z_{ik} = S)}{(\mu_B(\sum_k Z_{ik} = S))^N} \\
&= \frac{\prod_{k=1}^\infty \pi_k^{m_k}(1-\pi_k)^{N-m_k}}{\mathrm{PoiBin}(S|\{\pi_k\}_{k=1}^\infty)^N} \prod_{i=1}^N \mathbb{I}\left(\sum_{k=1}^\infty z_{ik} = S\right),
\end{aligned}
\tag{3}
$$

where $m_k = \sum_i z_{ik}$ and $\mathrm{PoiBin}(\cdot|\{\pi_k\}_{k=1}^\infty)$ is the infinite limit of the Poisson-binomial distribution [6], which describes the distribution over the number of successes in a sequence of independent but non-identical Bernoulli trials. The probability of $\mathbf{Z}$ given in Equation 3 is the infinite limit of the conditional Bernoulli distribution [6], which describes the distribution of the locations of the successes in such a trial, conditioned on their sum.

**Example 2** (Restriction of the iGaP to a fixed number of non-zero entries per row). The family of conditional distributions in the iGaP is given by the Poisson process, which has support in $\mathbb{N}^\infty$. Following Example 1, we can restrict this support to the set of all vectors $\mathbf{z} \in \mathbb{N}^\infty$ such that $\sum_k z_k \wedge 1 = S$ for some integer $S$ – i.e. the set of all non-negative integer-valued infinite vectors with $S$ non-zero entries. The conditional distribution of a matrix $\mathbf{Z} = \{\mathbf{z}_1, \ldots, \mathbf{z}_N\}$ under such a distribution is given by:

$$
\begin{aligned}
\mu_G^{|S}(Z = \mathbf{Z}) &= \frac{\prod_{i=1}^N \mu_G(Z_i = \mathbf{z}_i)\mathbb{I}(\sum_{k=1}^\infty z_{ik} \wedge 1 = S)}{(\mu_G(\sum_{k=1}^\infty Z_{ik} \wedge 1 = S))^N} \\
&= \frac{\prod_{k=1}^\infty \frac{\lambda_k^{m_k} e^{-\lambda_k}}{\prod_{i=1}^N z_{ik}!}}{\mathrm{PoiBin}(S|\{e^{-\lambda_k}\}_{k=1}^\infty)^N} \prod_{i=1}^N \mathbb{I}\left(\sum_{k=1}^\infty z_{ik} \wedge 1 = S\right).
\end{aligned}
\tag{4}
$$

**Example 3** (Restriction of the IBP to a random number of non-zero entries per row). Rather than specify the number of non-zero entries in each row a priori, we can allow it to be random, with some arbitrary distribution $f(\cdot)$ over the non-negative integers. A Bernoulli process restricted to have $f$-marginals can be described as

$$
\mu_B^{|f}(\mathbf{Z}) = \prod_{i=1}^N \mu_B^{|S_i}(Z_i = \mathbf{z}_i)f(S_i) = \prod_{i=1}^N \frac{f(S_i)\mathbb{I}(\sum_{k=1}^\infty z_{ik} = S_i)}{\mathrm{PoiBin}(S_i|\{\pi_k\}_{k=1}^\infty)} \prod_{k=1}^\infty \pi_k^{m_k}(1-\pi_k)^{N-m_k}, \tag{5}
$$

where $S_n = \sum_{k=1}^\infty z_{nk}$. If we marginalize over $B = \sum_{k=1}^\infty \pi_k \delta_{\phi_k}$, the resulting distribution is exchangeable, because mixtures of i.i.d. distributions are i.i.d.

We note that, even if we choose $f$ to be Poisson$(\alpha)$, we will not recover the IBP. The IBP has Poisson$(\alpha)$ marginals over the number non-zero elements per row, but the conditional distribution is described by a Poisson-binomial distribution. The Poisson-restricted IBP, however, will have Poisson marginal *and* conditional distributions.

Figure 1 shows some examples of samples from the single-parameter IBP, with parameter $\alpha = 5$, with various restrictions applied.

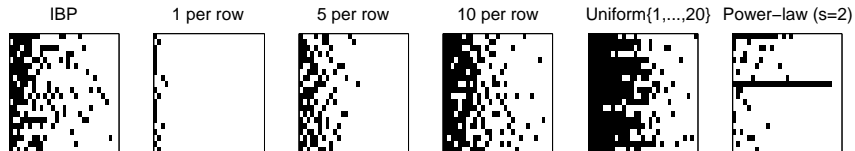

Figure 1: Samples from restricted IBPs.

## 3.2 Direct restriction of the predictive distributions

The construction in Section 3.1 is explicitly conditioned on a draw $B$ from the de Finetti mixing distribution $\nu$. Since it might be cumbersome to explicitly represent the infinite dimensional

object $B$, it is tempting to consider constructions that directly restrict the predictive distribution $p(X_{N+1}|X_1, \ldots, X_N)$, where $B$ has been marginalized out.

Unfortunately, the distribution over matrices obtained by this approach does not (in general – see the appendix for a counter-example) correspond to the distribution over matrices obtained by restricting the family of conditional distributions. Moreover, the resulting distribution will not in general be exchangeable. This means it is not appropriate for data sets where we have no explicit ordering of the data, and also means we cannot directly use the predictive distribution to define a Gibbs sampler (as is possible in exchangeable models).

**Theorem 2** (Sequences obtained by directly restricting the predictive distribution of an exchangeable sequence are not, in general, exchangeable)**.** *Let $p$ be the distribution of the unrestricted exchangeable model introduced in the proof of Theorem 1. Let $p^{*|A}$ be the distribution obtained by directly restricting this unrestricted exchangeable model such that $X_i \in A$, i.e.*

$$p^{*|A}(x_{N+1}|x_1, \ldots, x_N) \propto \mathbb{I}(x_{N+1} \in A) \frac{\int_\Theta \prod_{i=1}^{N+1} \mu_\theta(X = x_i)\nu(d\theta)}{\int_\Theta \prod_{i=1}^{N+1} \mu_\theta(X \in A)\nu(d\theta)} . \tag{6}$$

*In general, this will not be equal to Equation 2, and cannot be expressed as a mixture of i.i.d. distributions.*

*Proof.* To demonstrate that this is true, consider the counterexample given in Example 4. □

**Example 4** (A three-urn buffet)**.** Consider a simple form of the Indian buffet process, with a base measure consisting of three unit-mass atoms. We can represent the predictive distribution of such a model using three indexed urns, each containing one red ball (representing a one in the resulting matrix) and one blue ball (representing a zero in the resulting matrix). We generate a sequence of ball sequences by repeatedly picking a ball from each urn, noting the ordered sequence of colors, and returning the balls to their urns, plus one ball of each sampled color.

**Proposition 1.** *The three-urn buffet is exchangeable.*

*Proof.* By using the fact that a sequence is exchangeable iff the predictive distribution given the first $N$ elements of the sequence of the $N + 1$st and $N + 2$nd entries is exchangeable [9], it is trivial to show that this model is exchangeable and that, for example,

$$\begin{aligned} &p(X_{N+1} = (r, b, r), X_{N+2} = (r, r, b)|X_{1:N}) \\ =&\frac{m_1 m_2(N + 1 - m_3)}{(N + 1)^3} \cdot \frac{(m + 1 + 1)(N + 1 - m_2)m_3}{(N + 2)^3} \\ =&p(X_{N+1} = (r, r, b), X_{N+2} = (r, b, r)|X_{1:N}), \end{aligned} \tag{7}$$

where $m_i$ is the number of times in the first $N$ samples that the $i$th ball in a sample has been red. □

**Proposition 2.** *The directly restricted three-urn scheme (and, by extension, the directly restricted IBP) is not exchangeable.*

*Proof.* Consider the same scheme, but where the outcome is restricted such that there is one, and only one, red ball per sample. The probability of a sequence in this restricted model is given by

$$p^*(X_{N+1} = x|X_{1:N}) = \frac{\sum_{k=1}^3 \frac{m_k}{N+1-m_k} \mathbb{I}(x_i = r)}{\sum_{k=1}^3 \frac{m_k}{N+1-m_k}}$$

and, for example,

$$\begin{aligned} &p^*(X_{N+1} = (r, b, b), X_{N+2} = (b, r, b)|X_{1:N}) \\ =&\frac{\frac{m_1}{N+1-m_1}}{\sum_k \frac{m_k}{N+1-m_k}} \cdot \frac{\frac{m_2}{N+2-m_3}}{\frac{m_2}{N+1-m_2} - \frac{m_2}{N+2-m_2} + \sum_k \frac{m_k}{N+1-m_k}} \\ \neq&p^*(X_{N+1} = (b, r, b), X_{N+2} = (r, b, b)|X_{1:N}), \end{aligned} \tag{8}$$

therefore the restricted model is not exchangeable. By introducing a normalizing constant – corresponding to restricting over a subset of $\{0, 1\}^3$ – that depends on the previous samples, we have broken the exchangeability of the sequence.

By extension, a model obtained by directly restricting the predictive distribution of the IBP is not exchangeable. □

We note that there may well be situations where a non-exchangeable model, such as that described in Proposition 2, is appropriate for our data – for example where there is an explicit ordering on the data. It is not, however, an appropriate model if we believe our data to be exchangeable, or if we are interested in finding a single, stationary latent distribution describing our data. This exchangeable setting is the focus of this paper, so we defer exploration of distribution of non-exchangeable matrices obtained by restriction of the predictive distribution to future work.

## 4 Inference

We focus on models obtained by restricting the IBP to have $f$-marginals over the number of non-zero elements per row, as described in Example 3. Note that when $f = \delta_S$, this yields the setting described in Example 1. Extension to other cases, such as the restricted iGaP model of Example 2, are straightforward. We work with a truncated model, where we approximate the countably infinite sequence $\{\pi_k\}_{k=1}^{\infty}$ with a large, but finite, vector $\boldsymbol{\pi} := (\pi_1, \ldots, \pi_K)$, where each atom $\pi_k$ is distributed according to Beta$(\alpha/K, 1)$. An alternative approach would be to develop a slice sampler that uses a random truncation, avoiding the error introduced by the fixed truncation [14, 16]. We assume a likelihood function $g(\mathbf{X}|\mathbf{Z}) = \prod_i g(\mathbf{x}_i|\mathbf{z}_i)$.

### 4.1 Sampling the binary matrix Z

For marginal functions $f$ that assign probability mass to a contiguous, non-singleton subset of $\mathbb{N}$, we can Gibbs sample each entry of $\mathbf{Z}$ according to

$$p(z_{ik} = 1|\mathbf{x}_i, \boldsymbol{\pi}, \mathbf{Z}_{\neg ik}, \sum_{j \neq k} z_{ij} = a) \propto \pi_k \frac{f(a+1)}{p(\sum_k z_k = a+1|\boldsymbol{\pi})} g(x_i|z_{ik} = 1, \mathbf{Z}_{\neg ik})$$
$$p(z_{ik} = 0|\mathbf{x}_i, \boldsymbol{\pi}, \mathbf{Z}_{\neg ik}, \sum_{j \neq k} z_{ij} = a) \propto (1 - \pi_k) \frac{f(a)}{p(\sum_k z_k = a|\boldsymbol{\pi})} g(x_i|z_{ik} = 0, \mathbf{Z}_{\neg ik}).$$

(9)

Where $f = \delta_S$, this approach will fail, since any move that changes $z_{ik}$ must change $\sum_k z_{ik}$. In this setting, instead, we sample the locations of the non-zero entries $z_i^{(j)}$, $j = 1, \ldots, S$ of $\mathbf{z}_i$:

$$p(z_i^{(j)} = k|\mathbf{x}_i, \boldsymbol{\pi}, z_i^{(\neg j)}) \propto \pi_k(1 - \pi_k)^{-1} g(x_i|z_i^{(j)} = k, z_i^{(\neg j)}). \tag{10}$$

To improve mixing, we also include Metropolis-Hastings moves that propose an entire row of $\mathbf{Z}$. We include details in the supplementary material.

### 4.2 Sampling the beta process atoms $\pi$

Conditioned on $\mathbf{Z}$, the the distribution of $\boldsymbol{\pi}$ is

$$\nu(\{\pi_k\}_{k=1}^{\infty}|\mathbf{Z}) \propto \mu_{\{\pi_k\}}^{|f}(Z = \mathbf{Z})\nu(\{\pi_k\}_{k=1}^{\infty}) \propto \frac{\prod_{k=1}^{K} \pi_k^{(m_k + \frac{\alpha}{K} - 1)}(1 - \pi_k)^{N - m_k}}{\prod_{i=1}^{N} \text{PoiBin}(S_i|\boldsymbol{\pi})}. \tag{11}$$

The Poisson-binomial term can be calculated exactly in $O(K \sum_k z_{ik})$ using either a recursive algorithm [3, 5] or an algorithm based on the characteristic function that uses the Discrete Fourier Transform [8]. It can also be approximated using a skewed-normal approximation to the Poisson-binomial distribution [19]. We can therefore sample from the posterior of $\boldsymbol{\pi}$ using Metropolis Hastings steps. Since we believe our posterior will be close to the posterior for the unrestricted model, we use the proposal distribution $q(\pi_k|Z) = \text{Beta}(\alpha/K + m_k, N + 1 - m_k)$ to propose new values of $\pi_k$.

### 4.3 Evaluating the predictive distribution

In certain cases, we may wish to directly evaluate the predictive distribution $p^{|f}(\mathbf{z}_{N+1}|\mathbf{z}_1, \ldots, \mathbf{z}_N)$. Unfortunately, in the case of the IBP, we are unable to perform the integral in Equation 2 analytically. We can, however, *estimate* the predictive distribution using importance sampling. We sample $T$ measures $\boldsymbol{\pi}^{(t)} \sim \nu(\boldsymbol{\pi}|\mathbf{Z})$, where $\nu(\boldsymbol{\pi}|\mathbf{Z})$ is the posterior distribution over $\boldsymbol{\pi}$ in the finite approximation to the IBP, and then weight them to obtain the restricted predictive distribution

$$p^{|f}(\mathbf{z}_{N+1}|\mathbf{z}_1, \ldots, \mathbf{z}_N) \approx \frac{1}{T} \frac{\sum_{t=1}^{T} w_t \mu_{\boldsymbol{\pi}^{(t)}}^{|f}(\mathbf{z}_{N+1})}{\sum_t w_t}, \tag{12}$$

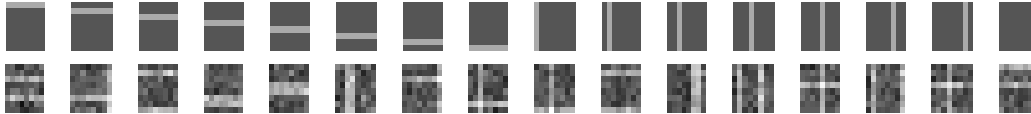

Figure 2: Top row: True features. Bottom row: Sample data points for $S = 2$.

| | $S = 2$ | $S = 5$ | $S = 8$ | $S = 11$ | $S = 14$ |
|---|---|---|---|---|---|
| IBP | $7297.4 \pm 2822.8$ | $8982.2 \pm 1981.7$ | $7442.8 \pm 3602.0$ | $8862.1 \pm 3920.2$ | $20244 \pm 6809.7$ |
| rIBP | $57.2 \pm 66.4$ | $3469.7 \pm 133.7$ | $5963.8 \pm 871.4$ | $11413 \pm 1992.9$ | $12199 \pm 2593.8$ |

Table 1: Structure error on synthetic data with 100 data points and $S$ features per data point.

where $w_t = \mu^{|f}_{\boldsymbol{\pi}^{(t)}}(\mathbf{z}_1, \ldots, \mathbf{z}_N)/\mu_{\boldsymbol{\pi}^{(t)}}(\mathbf{z}_1, \ldots, \mathbf{z}_N)$, and

$$\mu^{|f}_{\boldsymbol{\pi}}(\mathbf{Z}) \propto \prod_{i=1}^{N} \frac{f(S_i)\mathbf{I}(\sum_{k=1}^{K} z_{ik} = S_i)}{\text{PoiBin}(S_i|\boldsymbol{\pi})} \prod_{k=1}^{K} \pi_k^{m_k}(1 - \pi_k)^{N-m_k}.$$

## 5 Experimental evaluation

In this paper, we have described how distributions over exchangeable matrices, such as the IBP, can be modified to allow more flexible control over the distributions over the number of latent features. In this section, we perform experiments on both real and synthetic data. The synthetic data experiments are designed to show that appropriate restriction can yield more interpretable features. The experiments on real data are designed to show that careful choice of the distribution over the number of latent features in our models can lead to improved predictive performance.

### 5.1 Synthetic data

The IBP has been used to discover latent features that correspond to interpretable phenomena, such as latent causes behind patient symptoms [20]. If we have prior knowledge about the number of latent features per data point – for example, the number of players in a team, or the number of speakers in a conversation – we may expect both better predictive performance, and more interpretable latent features. In this experiment, we evaluate this hypothesis on synthetic data, where the true latent features are known. We generated images by randomly selecting $S$ of 16 binary features, shown in Figure 2, superimposing them, and adding isotropic Gaussian noise ($\sigma^2 = 0.25$). We modeled the resulting data using an uncollapsed linear Gaussian model, as described in [7], using both the IBP, and the IBP restricted to have $S$ features per row. To compare the generating matrix $\mathbf{Z}_0$ and our posterior estimate $\mathbf{Z}$, we looked at the structure error [20]. This is the sum absolute difference between the upper triangular portions of $\mathbf{Z}_0\mathbf{Z}_0^T$ and $\mathbb{E}[\mathbf{Z}\mathbf{Z}^T]$, and is a general measure of graph dissimilarity.

Table 1 shows the structure error obtained using both a standard IBP model (IBP) and an IBP restricted to have the correct number of latent features (rIBP), for varying numbers of features $S$. In each case, the number of data points is 100, the IBP parameter $\alpha$ is fixed to $S$, and the model is truncated to 50 features. Each experiment was repeated 10 times on independently generated data sets; we present the mean and standard deviation. All samplers were run for 5000 samples; the first 2500 were discarded as burn-in.

Where the number of features per data point is small relative to the total number of features, the restricted model does a much better job at recovering the "correct" latent structure. While the IBP may be able to explain the training data set as well as the restricted model, it will not in general recover the desired latent structure – which is important if we wish to interpret the latent structure.

Once the number of features per data point increases beyond half the total number of features, the model is ill-specified – it is more parsimonious to represent features via the *absence* of a bar. As a result, both models perform poorly at recovering the generating structure. The restricted model – and indeed the IBP – should only be expected to recover easily interpretable features where the number of such features per data point is small relative to the total number of features.

|      | 1     | 2     | 3     | 4     | 5     | 6     | 7     | 8     | 9     | 10    |
|------|-------|-------|-------|-------|-------|-------|-------|-------|-------|-------|
| IBP  | 0.591 | 0.726 | 0.796 | 0.848 | 0.878 | 0.905 | 0.923 | 0.936 | 0.952 | 0.958 |
| rIBP | 0.622 | 0.749 | 0.819 | 0.864 | 0.899 | 0.918 | 0.935 | 0.948 | 0.959 | 0.966 |
|      | 11    | 12    | 13    | 14    | 15    | 16    | 17    | 18    | 19    | 20    |
| IBP  | 0.961 | 0.969 | 0.974 | 0.978 | 0.982 | 0.989 | 0.991 | 0.996 | 0.997 | 1.000 |
| rIBP | 0.971 | 0.978 | 0.981 | 0.983 | 0.988 | 0.992 | 0.998 | 1.000 | 1.000 | 1.000 |

Table 2: Proportion correct at $n$ on classifying documents from the 20newsgroup data set.

## 5.2 Classification of text data

The IBP and its extensions have been used to directly model text data [17, 15]. In such settings, the IBP is used to directly model the presence or absence of words, and so the matrix $\mathbf{Z}$ is observed rather than latent, and the total number of features is given by the vocabulary size. We hypothesize that the Poisson assumption made by the IBP is not appropriate for text data, as the statistics of word use in natural language tends to follow a heavier tailed distribution [22]. To test this hypothesis, we modeled a collection of corpora using both an IBP, and an IBP restricted to have a negative Binomial distribution over the number of words. Our corpora were 20 collections of newsgroup postings on various topics (for example, comp.graphics, rec.autos, rec.sport.hockey)[1]. No pre-processing of the documents was performed. Since the vocabulary (and hence the feature space) is finite, we truncated both models to the vocabulary size. Due to the very large state space, we restricted our samples such that, in a single sample, atoms with the same posterior distribution were assigned the same value. For each model, $\alpha$ was set to the mean number of words per document in the corresponding group, and the maximum likelihood parameters were used for the negative Binomial distribution.

To evaluate the quality of the models, we classified held out documents based on their likelihood under each of the 20 newsgroups. This experiment is designed to replicate an experiment performed by [15] to compare the original and three-parameter IBP models. For both models, we estimated the predictive distribution by generating 1000 samples from the posterior of the beta process in the IBP model. For the IBP, we used these samples directly to estimate the predictive distribution; for the restricted model, we used the importance-weighted samples obtained using Equation 12. For each model, we trained on 1000 randomly selected documents, and tested on a further 1000 documents.

Table 2 shows the fraction of documents correctly classified in the first $n$ labels – i.e. the fraction of documents for which the correct labels is one of the $n$ most likely. The restricted IBP (rIBP) performs uniformly better than the unrestricted model.

## 6 Discussion and future work

The framework explored in this paper allows us to relax the distributional assumptions made by existing exchangeable nonparametric processes. As future work, we intend to explore which applications and models can most benefit from this greater flexibility.

We note that the model, as posed, suffers from an identifiability issue. Let $\tilde{B} = \sum_{k=1}^{\infty} \tilde{\pi}_k \delta_{\phi_k}$ be the measure obtained by transforming $B = \sum_{k=1}^{\infty} \pi_k \delta_{\phi_k}$ such that $\tilde{\pi}_k = \pi_k/(1 - \pi_k)$. Then, scaling $\tilde{B}$ by a positive scalar does not affect the likelihood of a given matrix $\mathbf{Z}$. We intend to explore the consequences of this in future work.

## Acknowledgments

We would like to thank Zoubin Ghahramani for valuable suggestions and discussions throughout this project. We would also like to thank Finale Doshi-Velez and Ryan Adams for pointing out the non-identifiability mentioned in Section 6. This research was supported in part by NSF grants DMS-1209194 and IIS-1111142, AFOSR grant FA95501010247, and NIH grant R01GM093156.

## Footnotes

[1]http://people.csail.mit.edu/jrennie/20Newsgroups/

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
