[Supplementary Material]

# Restricting exchangeable nonparametric distributions: Supplementary material

**Sinead A. Williamson**
University of Texas at Austin

**Steven N. MacEachern**
The Ohio State University

**Eric P. Xing**
Carnegie Mellon University

In this document, we give an example of a case where direct restriction of the predictive distribution of an exchangeable matrix yields a distribution over exchangeable matrices, and we give expanded details of the inference methods described in the main paper.

## 1 Direct restriction of the predictive distribution

Example 4 in the main paper gives an example of a situation where restricting the predictive distribution of an exchangeable model does not yield an exchangeable model. This example can be extended to general exchangeable matrices based on CRMs where we restrict the number of non-zero features, since we can interpret the probabilities of obtaining a non-zero entry as being described by a Bernoulli process. However, other forms of restriction can yield exchangeable matrices, as the following example shows:

**Example 5** (Restricting the row sums in the infinite gamma-Poisson process)**.** Consider restricting the predictive distribution of the infinite gamma-Poisson distribution such that each row sums to $S$. In the predictive distribution for the iGaP, for each previously observed feature $k$, we sample an element $X_{nk} \sim \text{NegBinom}(m_k, n/(n+1))$. We then sample a value $N_n^* \sim \text{NegBinom}(\theta, n/(n+1))$ and assign $N_n^*$ counts to new features according to a Chinese restaurant process. If we restrict this model such that each row sums to 1, we have:

$$p^{|1}(X_{(N+1)k} = 1|X_{1:N}) = \frac{p(X_{(N+1)k} = 1|X_{1:N}) \prod_{j \neq k} p(X_{(N+1)j} = 0|X_{1:N})}{p(\sum_j X_{(N+1)j} = 1|X_{1:N})}$$

$$= \begin{cases} \frac{m_k}{\sum_j m_j + \theta} & \text{if feature k has been seen before} \\ \frac{\theta}{\sum_j m_j + \theta} & \text{otherwise.} \end{cases}$$

In other words, the infinite gamma-Poisson process restricted to sum to one is a Chinese restaurant process. If we restrict the iGaP to sum to $S$, we have $S$ samples per data point from a Chinese restaurant process.

## 2 Metropolis Hastings proposals for Z

Our inference algorithm supplements the Gibbs sampling moves described by Equations 9 and 10 in the main paper with Metropolis Hastings steps. These steps allow us to propose larger moves, which can improve mixing. Let $\mathbf{z}_i^{(t)}$ be the value of the $i$th row of $\mathbf{Z}$ at iteration $t$. We propose a new row $\mathbf{z}^*$ by sampling from $\mu_{\{\pi_k\}}^f(\cdot)$ (e.g. by rejection sampling), and accept with probability

$$\min\left(1, \frac{g(\mathbf{x}_i|\mathbf{z}^*)}{g(\mathbf{x}_i|\mathbf{z}_i^{(t)})}\right).$$