[Reviews · NeurIPS 2013]

Submitted by Assigned_Reviewer_4

This is an interesting paper describing mechanism to introduce constraints of the priors

1) I have two concerns with how the discussion is formulated:
1.1) The introduction of the paper talks about preserving exchangeability, and a substantial effort is devoted to showing that restrictions on the predictive distributions do not necessarily lead to exchangeable rows. However, given that the authors are working with priors on array-valued processes, they should have framed the discussion in terms of separate exchangeability (Aldous, 1981), which is what we really desired from the model. I think that all the proofs works and the models described are indeed separately exchangeable (rather than unrestrictedly exchangeable), but the current presentation is not really adequate.

1.2) An implicit assumption is that the distribution on $\theta$ admits a density (denoted by $\nu$) with respect to some dominating measure. Besides not stating this clearly, the authors refer to $\nu$ as a measure in various parts of the manuscript.
Furthermore, the authors refer to $\theta$ as the "directing measure". $\theta$ is a collection of probabilities, but they do not add up to 1, so talking about a measure here seems incorrect. I think that precision in the use of standard terminology is important.

1.3) In equations (3) and (4), I believe that the authors should also multiply the final expressions by the appropriate indicator (as the support is now constrained)

2) Beyond being a mathematically interesting problem, the motivation for introducing the constraints is not very strong. The best motivation might come from the discussion on HMM models, but even that is relatively week. I guess that the authors could argue that obtaining better results in the experiment in Section 5.2 is motivation enough, but having some intuition on why the proposed model outperforms would have been useful.

3) There is no discussion about how the size of the truncation is to be selected. Now that constraints have been introduced, standard results will not be applicable (particularly if f(S) in equation (5) is heavy tailed).

4) Section 3.2 emphasizes that directly restricting the predictive distributions does not lead to an exchangeable model. The example is nice, but this is a well known property and I am not sure that it needed this much space.
Summary: A relatively straightforward paper without a very strong conceptual motivation.

Submitted by Assigned_Reviewer_6

The paper presents a novel exchangeable model based on restricting the Indian Buffet Process (IBP). This approach allows imposition of constraints on the number of latent features (for instance, they can be non-zero, exactly S, or power-law distributed). The paper presents a clear exposition of why and when exchangeability holds, a scheme for posterior inference, and results showing that the restricted model outperforms the IBP in simulations where the true number of features is known and on text where word distributions are known to be heavy-tailed.

In general, I found the paper to be both novel and interesting. I found the paper to be exceptionally clear and well written. The idea of restricted exchangeable distributions is one that has significant practical import (as demonstrated in the text example). The experiments were not as thorough as they could have been---for instance, it would have been nice to have an empirical case where the number of features was known---but it was sufficient to show advantages of the restricted approach over the IBP.
Summary: Interesting extension of known model that has significant practical import.

Submitted by Assigned_Reviewer_7

This paper provides a nice demonstration that restricting the support of de Finetti conditional distributions can give exchangeable models with many useful properties. Specifically, this paper cites de Finetti's law:
P(X_1, X_2...) = Int_Theta \prod mu_theta(X_i|theta)nu(theta)dtheta
and shows that restricting the support of the conditional distribution mu_theta can give rise to models with some particular constraints in mind. For example, by restricting an outcome from a Bernoulli process to contain exactly S "1"s, a version of the IBP where all rows sum to S is obtained.

Furthermore, the paper shows that directly restricting the predictive distribution p(x_n| x_1 .. x_{n-1}) does not (in general) give exchangeable distributions.

In the experiments the paper explores the IBP restricted to have f-marginals for several choices of f. For the IBP, the restricted conditionals lead to some drawbacks in inference: 1) the beta process atoms must be sampled (using a truncated representation in this case) 2) the predictive distribution can only be evaluated by approximate methods like importance sampling.

The experiments are simple but good. The paper shows an experiment using the IBP to directly model text data. The rIBP restricts the marginals on the number of words to be Negative Binomial, and improvements are seen over the IBP on the 20 newsgroups data.
Summary: This paper shows that restricting the conditional distribution appearing in the RHS of de Finetti's theorem can allow one to tailor exchangeable priors with desired marginal distributions. The experiments show the usefulness of such flexibility on a few simple examples.
Author Feedback

Author rebuttal: We thank the reviewers for their thoughtful comments.

Reviewer 4 asked for more emphasis on motivation. We recap our motivation here, and will increase the amount of space addressing this. In clustering applications, the practitioner has a number of choices of prior distributions that can allow him/her to capture properties of the data: If the number of features is known, he/she can use a Dirichlet distribution; if an exponential decrease is expected in cluster occupancy, a Dirichlet process; if a power-law decrease is expected, a Pitman-Yor or normalized generalized gamma process. Such choices can have significant effects on the predictive and descriptive power of models.

In a latent feature context, there is also a drive for flexibility in the distributional assumptions on the latent structure. The IBP has been used to find interpretable hidden causes for data, such as diseases underlying symptoms (Wood et al, 2006). There are many cases where we may have strong prior assumptions over the number of such causes. For example, when modeling the contributions of authors to papers, players to teams, or speakers to dialogues we may know the number of latent features per data point. When modeling preferences as bipartite graphs, we may expect a power law distribution over the number of edges, or a truncated number of edges. The IBP and its current extensions are not able to capture such a wide variety of distributional assumptions: The limitation of models such as the IBP to a Poisson number of non-zero entries has been noted in the literature (e.g. by Teh & Gorur, 2009), and only partially addressed (Caron (2012) allows a mixture of Poissons).

Our experiments bear out the intuition that, if we can craft a prior that better matches the underlying structure, we can better characterize the posterior with a finite number of observations, yielding better predictive power. In addition, the synthetic data experiments show that in cases where we know the posterior distribution is likely to be multimodal (e.g. due to identifiability issues), but where we have reasons to prefer interpretable modes, we can get more interpretable estimates by placing zero support on parameter settings known to be uninterpretable.


Reviewer 4 had several further questions that we address in turn.


1.1 - Separate exchangeability. We presented our discussion in terms of exchangeable vectors since this mirrors the approach taken in the existing IBP literature; however as Reviewer 4 rightly surmises, the construction presented yields separately exchangeable matrices. We are happy to incorporate discussion on the exchangeability of columns.


1.2 - Reviewer 4 is correct that we have overloaded notation for the distribution over theta and the density admitted by this distribution. We will adjust the notation to be in terms of measures rather than densities. Theta however is indeed a measure (just not a probability measure); the Bernoulli process is parameterized by an arbitrary discrete measure with atoms between zero and one but no constraints on the total mass. It is not a directing probability measure; however it implies a probability distribution over binary vectors. We will clarify this.

1.3 - The indicator variable should be included; we will change this for the final version.

Regarding Reviewer 4’s other comments, we will decrease the amount of space dedicated to section 3.2, in order to incorporate discussions of issues arising with truncation, and possible alternative approaches. We will also increase the amount of space devoted to motivation and discussion.

We thank Reviewers 6 and 7 for their positive comments. Since space is limited, we focused on straightforward, easily understood experiments; we will explore more complex models in later papers.

---

F. Caron. Bayesian nonparametric models for bipartite graphs. In NIPS, 2012.

Y. W. Teh and D. Gorur. Indian buffet processes with power law behaviour. In NIPS, 2009.

F. Wood, T. L. Griffiths, and Z. Ghahramani. A non-parametric Bayesian method for inferring hidden causes. In UAI, 2006.